# The Strong Screening Rule for SLOPE

**Johan Larsson**
Dept. of Statistics, Lund University
`johan.larsson@stat.lu.se`

**Małgorzata Bogdan**
Dept. of Mathematics, University of Wroclaw
Dept. of Statistics, Lund University
`malgorzata.bogdan@uwr.edu.pl`

**Jonas Wallin**
Dept. of Statistics, Lund University
`jonas.wallin@stat.lu.se`

## Abstract

Extracting relevant features from data sets where the number of observations ($n$) is much smaller then the number of predictors ($p$) is a major challenge in modern statistics. Sorted L-One Penalized Estimation (SLOPE)—a generalization of the lasso—is a promising method within this setting. Current numerical procedures for SLOPE, however, lack the efficiency that respective tools for the lasso enjoy, particularly in the context of estimating a complete regularization path. A key component in the efficiency of the lasso is predictor screening rules: rules that allow predictors to be discarded before estimating the model. This is the first paper to establish such a rule for SLOPE. We develop a screening rule for SLOPE by examining its subdifferential and show that this rule is a generalization of the strong rule for the lasso. Our rule is heuristic, which means that it may discard predictors erroneously. In our paper, however, we show that such situations are rare and easily safeguarded against by a simple check of the optimality conditions. Our numerical experiments show that the rule performs well in practice, leading to improvements by orders of magnitude for data in the $p \gg n$ domain, as well as incurring no additional computational overhead when $n > p$.

## 1 Introduction

Extracting relevant features from data sets where the number of observations ($n$) is much smaller then the number of predictors ($p$) is one of the major challenges in modern statistics. The dominating method for this problem, in a regression setting, is the lasso [1]. Recently, however, an alternative known as Sorted L-One Penalized Estimation (SLOPE) has been proposed [2–4], which is a generalization of the Octagonal Shrinkage and Supervised Clustering Algorithm for Regression (OSCAR) [5].

SLOPE is a regularization method that uses the sorted $\ell_1$ norm instead of the regular $\ell_1$ norm, which is used in the lasso. SLOPE features several interesting properties, such as control of the false discovery rate [2, 6], asymptotic minimaxity [7], and clustering of regression coefficients in the presence of strong dependence between predictors [8].

In more detail, SLOPE solves the convex optimization problem

$$\text{minimize}_{\beta \in \mathbb{R}^p} \{f(\beta) + J(\beta; \lambda)\}, \tag{1}$$

where $f(\beta)$ is smooth and convex and $J(\beta; \lambda) = \sum_{j=1}^{p} \lambda_j |\beta|_{(j)}$ is the convex but non-smooth sorted $\ell_1$ norm [2, 8], where $|\beta|_{(1)} \geq |\beta|_{(2)} \geq \cdots \geq |\beta|_{(p)}$ and $\lambda_1 \geq \cdots \geq \lambda_p \geq 0$.

It is easy to see that the lasso is a specific instance of SLOPE, obtained by setting all elements of $\lambda$ to the same value. But in contrast to SLOPE based on a decreasing sequence $\lambda$, the lasso suffers from unpredictable behavior in the presence of highly correlated predictors, occasionally resulting in solutions wherein only a subset among a group of correlated predictors is selected. SLOPE, in contrast, turns out to be robust to correlated designs, which it accomplishes via clustering: setting coefficients of predictors to the same value [8]. Kremer et al. [9] showed that this clustering is related to similarity of the influence of respective variables on the likelihood function, which can occur due to strong correlation [8, 10] but also due to similarity of true regression coefficients [11]. This property in some cases allows SLOPE to select all $p$ predictors if they are grouped into no more than $n$ clusters [9, 11], while the lasso can at most select $n$ predictors [12].

The choice of $\lambda$ sequence in (1) typically needs to be based on cross-validation or similar schemes. Most algorithms for fitting sparse regression, such as as the one implemented for lasso in the `glmnet` package for R [13], accomplish this by constructing a path of decreasing $\lambda$. For SLOPE, we begin the path with $\lambda^{(1)}$ and finish at $\lambda^{(l)}$ with $\lambda_j^{(m)} \geq \lambda_j^{(m+1)}$ for $j = 1, 2, \ldots, p$ and $m = 1, 2, \ldots, l-1$. (See Section 3.1 for details regarding the construction of the path.) For any point along this path, we let $\hat{\beta}(\lambda^{(m)})$ be the respective SLOPE estimate, such that

$$\hat{\beta}(\lambda^{(m)}) = \arg\min_{\beta \in \mathbb{R}^p} \left\{ f(\beta) + J(\beta; \lambda^{(m)}) \right\}.$$

Fitting the path repeatedly by cross-validation introduces a heavy computational burden. For the lasso, an important remedy for this issue arose with the advent of screening rules, which provide criteria for discarding predictors before fitting the model.

Screening rules can be broken down into two categories: *safe* and *heuristic* (unsafe) screening rules. The former of these guarantee that any predictors screened as inactive (determined to be zero by the rule) are in fact zero at the solution. Heuristic rules, on the other hand, may lead to *violations:* incorrectly discarding predictors, which means that heuristic rules must be supplemented with a check of the Karush–Kuhn–Tucker (KKT) conditions. For any predictors failing the test, the model must be refit with these predictors added back in order to ensure optimality.

Safe screening rules include the safe feature elimination rule (SAFE [14]), the dome test [15], Enhanced Dual-Polytope Projection (EDPP [16]), and the Gap Safe rule [17, 18]. Heuristic rules include Sure Independence Screening (SIS [19]), Blitz [20], and the strong rule [21]. There have also been attempts to design *dynamic* approaches to screening [22] as well as hybrid rules, utilizing both heuristic and safe rules in tandem [23].

The implications of screening rules have been remarkable, allowing lasso models in the $p \gg n$ domain to be solved in a fraction of the time required otherwise and with a much reduced memory footprint [21]. Implementing screening rules for SLOPE has, however, proven to be considerably more challenging. After the first version of this paper appeared on *arXiv* [24], a first safe rule for SLOPE has been published [25]. Yet, because of the non-separability of the penalty in SLOPE, this rule requires iterative screening during optimization, which means that predictors cannot be screened prior to fitting the model. This highlights the difficulty in developing screening rules for SLOPE.

Our main contribution in this paper is the presentation of a first heuristic screening rule for SLOPE based on the strong rule for the lasso. In doing so, we also introduce a novel formulation of the subdifferential for the sorted $\ell_1$ norm. We then proceed to show that this rule is effective, rarely leads to violations, and offers performance gains comparable to the strong rule for the lasso.

## 1.1 Notation

We use uppercase letters for matrices and lowercase letters for vectors and scalars. $\mathbf{1}$ and $\mathbf{0}$ denote vectors with all elements equal to 1 and 0 respectively, with dimension inferred from context. We use $\prec$ and $\succ$ to denote element-wise relational operators. We also let $\mathrm{card}\,\mathcal{A}$ denote the cardinality of set $\mathcal{A}$ and define $\mathrm{sign}\,x$ to be the signum function with range $\{-1, 0, 1\}$. Furthermore, we define $x_\downarrow$ to refer to a version of $x$ sorted in decreasing order and the cumulative sum function for a vector $x \in \mathbb{R}^n$ as $\mathrm{cumsum}(x) = [x_1, x_1 + x_2, \cdots, \sum_{i=1}^{\tilde{n}} x_i]^T$. We also let $|i|$ be the index operator of $y \in \mathbb{R}^p$ so that $|y_{|i|}| = |y|_{(i)}$ for all $i = 1, \ldots, p$. Finally, we allow a vector to be indexed with an integer-valued set by eliminating those elements of this vector whose indices do not belong to the indexing set—for instance, if $\mathcal{A} = \{3, 1\}$ and $v = [v_1, v_2, v_3]^T$, then $v_\mathcal{A} = [v_1, v_3]^T$.

## 2 Theory

Proofs of the following theorem and propositions are provided in the supplementary material.

### 2.1 The Subdifferential for SLOPE

The basis of the strong rule for $\ell_1$-regularized models is the subdifferential. By the same argument, we now turn to the subdifferential of SLOPE. The subdifferential for SLOPE has been derived previously as a characterization based on polytope mappings [26, 27]; here we present an alternative formulation that can be used as the basis of an efficient algorithm. First, however, let $\mathcal{A}_i(\beta) \subseteq \{1, \ldots, p\}$ denote a set of indices for $\beta \in \mathbb{R}^p$ such that

$$\mathcal{A}_i(\beta) = \{j \in \{1, \ldots, p\} \mid |\beta_i| = |\beta_j|\} \tag{2}$$

where $\mathcal{A}_i(\beta) \cap \mathcal{A}_l(\beta) = \varnothing$ if $l \notin \mathcal{A}_i(\beta)$. To keep notation concise, we let $\mathcal{A}_i$ serve as a shorthand for $\mathcal{A}_i(\beta)$. In addition, we define the operator $O : \mathbb{R}^p \to \mathbb{N}^p$, which returns a permutation that rearranges its argument in descending order by its absolute values and $R : \mathbb{R}^p \to \mathbb{N}^p$, which returns the ranks of the absolute values in its argument.

**Example 1.** *If $\beta = \{-3, 5, 3, 6\}$, then $\mathcal{A}_1 = \{1, 3\}$, $O(\beta) = \{4, 2, 1, 3\}$, and $R(\beta) = \{3, 2, 4, 1\}$.*

**Theorem 1.** *The subdifferential $\partial J(\beta; \lambda) \in \mathbb{R}^p$ is the set of all $g \in \mathbb{R}^p$ such that*

$$g_{\mathcal{A}_i} = \left\{ s \in \mathbb{R}^{\operatorname{card} \mathcal{A}_i} \, \middle| \, \begin{cases} \operatorname{cumsum}(|s|_{\downarrow} - \lambda_{R(s)_{\mathcal{A}_i}}) \preceq \mathbf{0} & \text{if } \beta_{\mathcal{A}_i} = \mathbf{0}, \\ \operatorname{cumsum}(|s|_{\downarrow} - \lambda_{R(s)_{\mathcal{A}_i}}) \preceq \mathbf{0} \\ \quad \text{and } \sum_{j \in \mathcal{A}_i}\left(|s_j| - \lambda_{R(s)_j}\right) = 0 & \text{otherwise.} \end{cases} \right\}$$

### 2.2 Screening Rule for SLOPE

#### 2.2.1 Sparsity Pattern

Recall that we are attempting to solve the following problem: we know $\hat{\beta}(\lambda^{(m)})$ and want to predict the support of $\hat{\beta}(\lambda^{(m+1)})$, where $\lambda^{(m+1)} \preceq \lambda^{(m)}$. The KKT stationarity criterion for SLOPE is

$$\mathbf{0} \in \nabla f(\beta) + \partial J(\beta; \lambda), \tag{3}$$

where $\partial J(\beta; \lambda)$ is the subdifferential for SLOPE (Theorem 1). This means that if $\nabla f(\hat{\beta}(\lambda^{(m+1)}))$ was available to us, we could identify the support exactly. In Algorithm 1, we present an algorithm to accomplish this in practice.

| **Algorithm 1** | **Algorithm 2** Fast version of Algorithm 1. |
|---|---|
| **Require:** $c \in \mathbb{R}^p$, $\lambda \in \mathbb{R}^p$, where $\lambda_1 \geq \cdots \geq \lambda_p \geq 0$. | **Require:** $c \in \mathbb{R}^p$, $\lambda \in \mathbb{R}^p$, where $\lambda_1 \geq \cdots \geq \lambda_p \geq 0$ |
| 1: $\mathcal{S}, \mathcal{B} \leftarrow \varnothing$ | 1: $i \leftarrow 1$, $k \leftarrow 0$, $s \leftarrow 0$ |
| 2: **for** $i \leftarrow 1, \ldots, p$ **do** | 2: **while** $i + k \leq p$ **do** |
| 3: $\quad \mathcal{B} \leftarrow \mathcal{B} \cup \{i\}$ | 3: $\quad s \leftarrow s + c_{i+k} - \lambda_{i+k}$ |
| 4: $\quad$ **if** $\sum_{j \in \mathcal{B}}\left(c_j - \lambda_j\right) \geq 0$ **then** | 4: $\quad$ **if** $s \geq 0$ **then** |
| 5: $\quad\quad \mathcal{S} \leftarrow \mathcal{S} \cup \mathcal{B}$ | 5: $\quad\quad k \leftarrow k + i$ |
| 6: $\quad\quad \mathcal{B} \leftarrow \varnothing$ | 6: $\quad\quad i \leftarrow 1$ |
| 7: $\quad$ **end if** | 7: $\quad\quad s \leftarrow 0$ |
| 8: **end for** | 8: $\quad$ **else** |
| 9: **return** $\mathcal{S}$ | 9: $\quad\quad i \leftarrow i + 1$ |
| | 10: $\quad$ **end if** |
| | 11: **end while** |
| | 12: **return** $k$ |

In Proposition 1, we show that the result of Algorithm 1 with $c := |\nabla f(\hat{\beta}(\lambda^{(m+1)}))|_{\downarrow}$ and $\lambda := \lambda^{(m+1)}$ as input is certified to contain the true support set of $\hat{\beta}(\lambda^{(m+1)})$.

**Proposition 1.** *Taking $c := |\nabla f(\hat{\beta}(\lambda^{(m+1)}))|_{\downarrow}$ and $\lambda := \lambda^{(m+1)}$ as input to Algorithm 1 returns a superset of the true support set of $\hat{\beta}(\lambda^{(m+1)})$.*

**Remark 1.** *In Algorithm 1, we implicitly make use of the fact that the results are invariant to permutation changes within each cluster $\dot{\mathcal{A}}_i$ (as defined in (2))—a fact that follows directly from the definition of the subdifferential (Theorem 1). In particular, this means that the indices for the set of inactive predictors will be ordered last in both $|\hat{\beta}|_{\downarrow}$ and $|\nabla f(\hat{\beta})|_{\downarrow}$; that is, for all $i, j \in \{1, 2, \ldots, p\}$ such that $\hat{\beta}_i = 0$, $\hat{\beta}_j \neq 0$,*

$$O(\nabla f(\hat{\beta}))_i > O(\nabla f(\hat{\beta}))_j \implies O(\hat{\beta})_i > O(\hat{\beta})_j,$$

*which allows us to determine the sparsity in $\hat{\beta}$ via $\nabla f(\hat{\beta})$.*

Proposition 1 implies that Algorithm 1 may lead to a conservative decision by potentially including some of the support of inactive predictors in the result, i.e. indices for which the corresponding coefficients are in fact zero. To see this, let $\mathcal{U} = \{l, l+1, \ldots, p\}$ be a set of inactive predictors and take $c := |\nabla f(\hat{\beta}(\lambda^{(m+1)}))|_{\downarrow}$. For every $k \in \mathcal{U}$, $k \geq l$ for which $\sum_{i=l}^{k}(c_i - \lambda_i) = 0$, $\{l, l+1, \ldots, k\}$ will be in the result of Algorithm 1 in spite of being inactive. This situation, however, occurs only when $c$ is the true gradient at the solution and for this reason is of little practical importance.

Since the check in Algorithm 1 hinges only on the last element of the cumulative sum at any given time, we need only to store and update a single scalar instead of the full cumulative sum vector. Using this fact, we can derive a fast version of the rule (Algorithm 2), which returns $k$: the predicted number of active predictors at the solution.[1]

Since we only have to take a single pass over the predictors, the cost of the algorithm is linear in $p$. To use the algorithm in practice, however, we first need to compute the gradient at the previous solution and sort it. Using least squares regression as an example, this results in a complexity of $\mathcal{O}(np + p \log p)$. To put this into perspective, this is (slightly) lower than the cost of a single gradient step if a first-order method is used to compute the SLOPE solution (since it also requires evaluation of the proximal operator).

### 2.2.2 Gradient Approximation

The validity of Algorithm 1 requires $\nabla f(\hat{\beta}(\lambda^{(m+1)}))$ to be available, which of course is not the case. Assume, however, that we are given a reasonably accurate surrogate of the gradient vector and suppose that we substitute this estimate for $\nabla f(\hat{\beta}(\lambda^{(m+1)}))$ in Algorithm 1. Intuitively, this should yield us an estimate of the active set—the better the approximation, the more closely this screened set should resemble the active set. For the sequel, let $\mathcal{S}$ and $\mathcal{T}$ be the screened and active set respectively.

An obvious consequence of using our approximation is that we run the risk of picking $\mathcal{S} \not\supseteq \mathcal{T}$, which we then naturally must safeguard against. Fortunately, doing so requires only a KKT stationarity check—whenever the check fails, we relax $\mathcal{S}$ and refit. If such failures are rare, it is not hard to imagine that the benefits of tackling the reduced problem might outweigh the costs of these occasional failures.

Based on this argument, we are now ready to state the strong rule for SLOPE, which is a natural extension of the strong rule for the lasso [21]. Let $\mathcal{S}$ be the output from running Algorithm 1 with

$$c := \left(|\nabla f(\hat{\beta}(\lambda^{(m)}))| + \lambda^{(m)} - \lambda^{(m+1)}\right)_{\downarrow}, \qquad \lambda := \lambda^{(m+1)}$$

as input. The strong rule for SLOPE then discards all predictors corresponding to $\mathcal{S}^c$.

**Proposition 2.** *Let $c_j(\lambda) = (\nabla f(\hat{\beta}(\lambda)))_{|j|}$. If $|c_j'(\lambda)| \leq 1$ for all $j = 1, 2, \ldots, p$ and $O(c(\lambda^{(m+1)})) = O(c(\lambda^{(m)}))$ (see Section 2.1 for the definition of $O$), the strong rule for SLOPE returns a superset of the true support set.*

Except for the assumption on fixed ordering permutation, the proof for Proposition 2 is comparable to the proof of the strong rule for the lasso [21]. The bound appearing in the proposition, $|c_j'(\lambda)| \leq 1$, is referred to as the *unit slope bound*, which results in the following rule for the lasso: discard the $j$th predictor if

$$\left|\nabla f(\beta(\lambda^{(m)}))_j\right| \leq 2\lambda_j^{(m+1)} - \lambda_j^{(m)}.$$

In Proposition 3, we formalize the connection between the strong rule for SLOPE and lasso.

**Proposition 3.** *The strong rule for SLOPE is a generalization of the strong rule for the lasso; that is, when $\lambda_j = \lambda_i$ for all $i, j \in \{1, \ldots, p\}$, the two rules always produce the same screened set.*

Finally, note that a non-sequential (basic) version of this rule is obtained by simply using the gradient for the null model as the basis for the approximation together with the penalty sequence corresponding to the point at which the first predictor enters the model (see Section 3.1).

### 2.2.3 Violations of the Rule

Violations of the strong rule for SLOPE occur only when the unit slope bound fails, which may happen for both inactive and active predictors—in the latter case, this can occur when the clustering or the ordering permutation changes for these predictors. This means that the conditions under which violations may arise for the strong rule for SLOPE differ from those corresponding to the strong rule for the lasso [21].

To safeguard against violations, we check the KKT conditions after each fit and add violating predictors to the screened set, refit, and repeat the checks until there are no violations. In Section 3.2.2, we will study the prevalence of violations in simulated experiments.

### 2.2.4 Algorithms

Tibshirani et al. [21] considered two algorithms using the strong rule for the lasso. In this paper, we consider two algorithms that are analogous except in one regard. First, however, let $\mathcal{S}(\lambda)$ be the strong set, i.e. the set obtained by application of the strong rule for SLOPE, and $\mathcal{T}(\lambda)$ the active set. Both algorithms begin with a set $\mathcal{E}$ of predictors, fit the model to this set, and then either expand this set, refit and repeat, or stop.

In the *strong set* algorithm (see supplementary material for details) we initialize $\mathcal{E}$ with the union of the strong set and the set of predictors active at the previous step on the regularization path. We then fit the model and check for KKT violations in the full set of predictors, expanding $\mathcal{E}$ to include any predictors for which violations occur and repeat until there are no violations.

In the *previous set* algorithm (see supplementary material for details) we initialize $\mathcal{E}$ with only the set of previously active predictors, fit, and check the KKT conditions against the strong rule set. If there are violations in the strong set, the corresponding predictors are added to $\mathcal{E}$ and the model is refit. Only when there are no violations in the strong set do we check the KKT conditions in the full set. This procedure is repeated until there are no violations in the full set.

These two algorithms differ from the strong and working set algorithms from Tibshirani et al. [21] in that we use only the set of previously active predictors rather than the set of predictors that have been active at any previous step on the path.

## 3 Experiments

In this section we present simulations that examine the effects of applying the screening rules. The problems here reflect our focus on problems in the $p \gg n$ domain, but we will also briefly consider the reverse in order to examine the potential overhead of the rules when $n > p$.

### 3.1 Setup

Unless stated otherwise, we will use the strong set algorithm with the strong set computed using Algorithm 2. Unless stated otherwise, we normalize the predictors such that $\bar{x}_j = 0$ and $\|x_j\|_2 = 1$ for $j = 1, \ldots, p$. In addition, we center the response vector such that $\bar{y} = 0$ when $f(\beta)$ is the least squares objective.

We use the *Benjamini–Hochberg* (BH) method [3] for computing the sequence, which sets $\lambda_i^{\mathrm{BH}} = \Phi^{-1}\big(1 - qi/(2p)\big)$ for $i = 1, 2, \ldots, p$, where $\Phi^{-1}$ is the probit function.[2] To construct the regularization path, we parameterize the sorted $\ell_1$ penalty as $J(\beta; \lambda, \sigma) = \sigma \sum_{j=1}^{p} |\beta|_{(j)} \lambda_j$, with

$\sigma^{(1)} > \sigma^{(2)} > \cdots > \sigma^{(l)} > 0$. We pick $\sigma^{(1)}$ corresponding to the point at which the first predictor enters the model, which corresponds to maximizing $\sigma \in \mathbb{R}$ subject to $\mathrm{cumsum}(\nabla f(\mathbf{0})_{\downarrow} - \sigma\lambda) \preceq 0$, which is given explicitly as

$$\sigma^{(1)} = \max(\mathrm{cumsum}(\nabla f(\mathbf{0})_{\downarrow}) \oslash \mathrm{cumsum}(\lambda)),$$

where $\oslash$ is the Hadamard (element-wise) division operator. We choose $\sigma^{(l)}$ to be $t\sigma^{(1)}$ with $t = 10^{-2}$ if $n < p$ and $10^{-4}$ otherwise. Unless stated otherwise, we employ a regularization path of $l = 100$ $\lambda$ sequences but stop this path prematurely if 1) the number of unique coefficient magnitudes exceed the number of observations, 2) the fractional change in deviance from one step to another is less than $10^{-5}$, or 3) if the fraction of deviance explained exceeds $0.995$.

Throughout the paper we use version 0.2.1 of the R package `SLOPE` [28], which uses the accelerated proximal gradient algorithm FISTA [29] to estimate all models; convergence is obtained when the duality gap as a fraction of the primal and the relative level of infeasibility [30] are lower than $10^{-5}$ and $10^{-3}$ respectively. All simulations were run on a dedicated high-performance computing cluster and the code for the simulations is available in the supplementary material and at `https://github.com/jolars/slope-screening-code/`.

### 3.2 Simulated Data

Let $X \in \mathbb{R}^{n \times p}$, $\beta \in \mathbb{R}^{p \times m}$, and $y \in \mathbb{R}^n$. We take

$$y_i = x_i^T \beta + \varepsilon_i, \qquad i = 1, 2, \ldots, n,$$

where $\varepsilon_i$ are sampled from independently and identically distributed standard normal variables. $X$ is generated such that each row is sampled independently and identically from a multivariate normal distribution $\mathcal{N}(\mathbf{0}, \Sigma)$. From here on out, we also let $k$ denote the cardinality of the non-zero support set of the true coefficients, that is, $k = \mathrm{card}\{i \in \mathbb{N}^p \mid \beta_i \neq 0\}$.

#### 3.2.1 Efficiency

We begin by studying the efficiency of the strong rule for SLOPE on problems with varying levels of correlation $\rho$. Here, we let $n = 200$, $p = 5000$, and $\Sigma_{ij} = 1$ if $i = j$ and $\rho$ otherwise. We take $k = p/4$ and generate $\beta_i$ for $i = 1, \ldots, k$ from $\mathcal{N}(0, 1)$. We then fit a least squares regression model regularized with the sorted $\ell_1$ norm to this data and screen the predictors with the strong rule for SLOPE. Here we set $q = 0.005$ in the construction of the BH sequence.

The size of the screened set is clearly small next to the full set (Figure 1). Note, however, that the presence of strong correlation among the predictors both means that there is less to be gained by screening since many more predictors are active at the start of the path, as well as makes the rule more conservative. No violations of the rule were observed in these simulations.

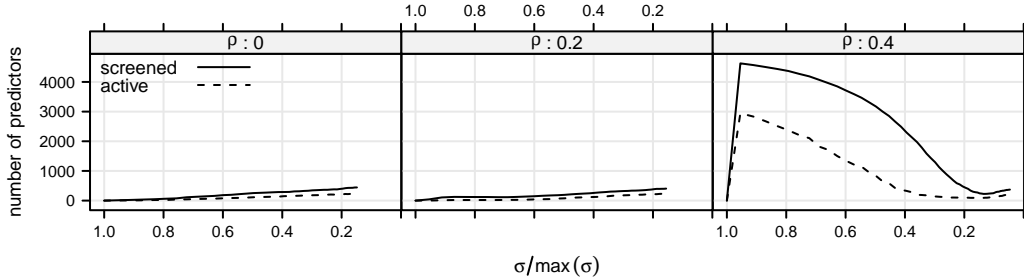

Figure 1: Number of screened and active predictors for sorted $\ell_1$-regularized least squares regression using no screening or the strong rule for SLOPE.

#### 3.2.2 Violations

To examine the number of violations of the rule, we generate a number of data sets with $n = 100$, $p \in \{20, 50, 100, 500, 1000\}$, and $\rho = 0.5$. We then fit a full path of 100 $\lambda$ sequences across 100 iterations, averaging the results. (Here we disable the rules for prematurely aborting the path described

at the start of this section.) We sample the first fourth of the elements of $\beta$ from $\{-2, 2\}$ and set the rest to zero.

Violations appear to be rare in this setting and occur only for the lower range of $p$ values (Figure 2). For $p = 100$, for instance, we would at an average need to estimate roughly 100 paths for this type of design to encounter a single violation. Given that a complete path consists of 100 steps and that the warm start after the violation is likely a good initialization, this can be considered a marginal cost.

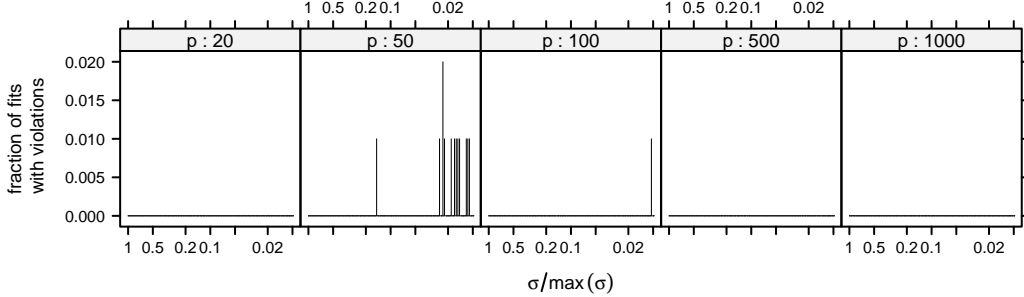

Figure 2: Fraction of model fits resulting in violations of the strong rule for sorted $\ell_1$-regularized least squares regression.

### 3.2.3 Performance

In this section, we study the performance of the screening rule for sorted $\ell_1$-penalized least squares, logistic, multinomial, and Poisson regression.

We now take $p = 20,000$, $n = 200$, and $k = 20$. To construct $X$, we let $X_1, X_2, \ldots, X_p$ be random variables distributed according to

$$X_1 \sim \mathcal{N}(\mathbf{0}, I), \qquad X_j \sim \mathcal{N}(\rho X_{j-1}, I) \quad \text{for } j = 2, 3, \ldots, p,$$

and sample the $j$th column in $X$ from $X_j$ for $j = 1, 2, \ldots, p$.

For least squares and logistic regression data we sample the first $k = 20$ elements of $\beta$ without replacement from $\{1, 2, \ldots, 20\}$. Then we let $y = X\beta + \varepsilon$ for least squares regression and $y = \text{sign}(X\beta + \varepsilon)$ for logistic regression, in both cases taking $\varepsilon \sim \mathcal{N}(\mathbf{0}, 20I)$. For Poisson regression, we generate $\beta$ by taking random samples without replacement from $\{\frac{1}{40}, \frac{2}{40}, \ldots, \frac{20}{40}\}$ for its first 20 elements. Then we sample $y_i$ from $\text{Poisson}\left(\exp((X\beta)_i)\right)$ for $i = 1, 2, \ldots, n$. For multinomial regression, we start by taking $\beta \in \mathbb{R}^{p \times 3}$, initializing all elements to zero. Then, for each row in $\beta$ we take a random sample from $\{1, 2, \ldots, 20\}$ without replacement and insert it at random into one of the elements of that row. Then we sample $y_i$ randomly from $\text{Categorical}(3, p_i)$ for $i = 1, 2, \ldots, n$, where

$$p_{i,l} = \frac{\exp\left((X\beta)_{i,l}\right)}{\sum_{l=1}^{3} \exp\left((X\beta)_{i,l}\right)}.$$

The benchmarks reveal a strong effect on account of the screening rule through the range of models used (Figure 3), leading to a substantial reduction in run time. As an example, the run time for fitting logistic regression when $\rho = 0.5$ decreases from roughly 70 to 5 seconds when the screening rule is used.

We finish this section with an examination of two types of algorithms outlined in Section 2.2.4: the strong set and previous set algorithm. In Figure 1 we observed that the strong rule is conservative when correlation is high among predictors, which indicates that the previous set algorithm might yield an improvement over the strong set algorithm.

In order to examine this, we conduct a simulation in which we vary the strength of correlation between predictors as well as the parameter $q$ in the construction of the BH regularization sequence. Motivation for varying the latter comes from the relationship between coefficient clustering and the intervals in the regularization sequence—higher values of $q$ cause larger gaps in the sequence, which in turn leads to more clustering among predictors. This clustering, in turn, is strongest at the start of the path when regularization is strong.

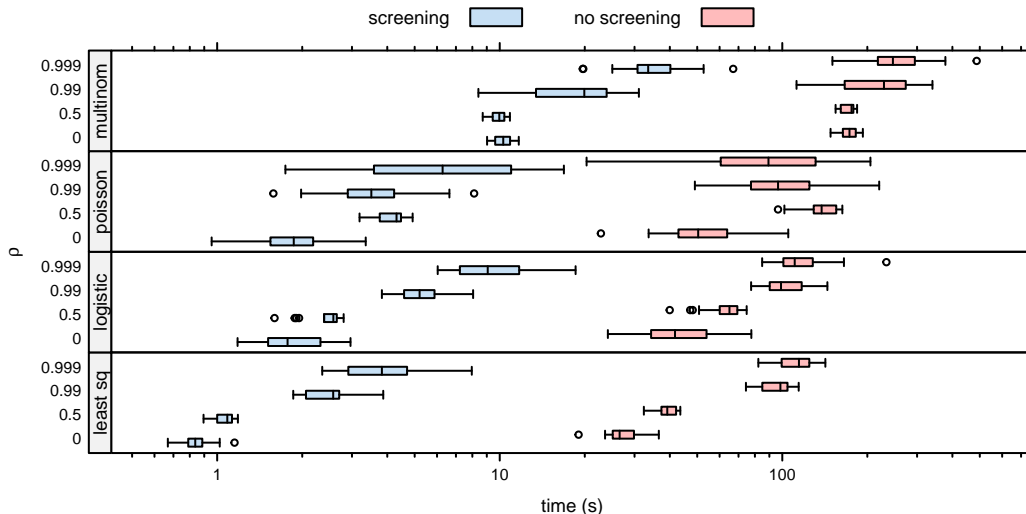

Figure 3: Time taken to fit SLOPE with or without the strong screening rule for randomly generated data.

For large enough $q$ and $\rho$, this behavior in fact occasionally causes almost all predictors to enter the model at the second step on the path. As an example, using when $\rho = 0.6$ and fitting with $q = 10^{-2}$ and $10^{-4}$ leads to 2215 and 8 nonzero coefficients respectively at the second step in one simulation.

Here, we let $n = 200$, $p = 5000$, $k = 50$, and $\rho \in \{0, 0.1, 0.2, \ldots, 0.8\}$. The data generation process corresponds to the setup at the start of this section for least squares regression data except for the covariance structure of $X$, which is equal to that in Section 3.2.1. We sample the non-zero entries in $\beta$ independently from a random variable $U \sim \mathcal{N}(0, 1)$.

The two algorithms perform similarly for $\rho \le 0.6$ (Figure 4). For larger $\rho$, the previous set strategy evidently outperforms the strong set strategy. This result is not surprising: consider Figure 1, for instance, which shows that the behavior of the regularization path under strong correlation makes the previous set strategy particularly effective in this context.

## 3.3 Real Data

### 3.3.1 Efficiency and Violations

We examine efficiency and violations for four real data sets: *arcene*, *dorothea*, *gisette*, and *golub*, which are the same data sets that were examined in Tibshirani et al. [21]. The first three originate from Guyon et al. [31] and were originally collected from the UCI (University of California Irvine) Machine Learning Repository [32], whereas the last data set, *golub*, was originally published in Golub et al. [33]. All of the data sets were collected from `http://statweb.stanford.edu/~tibs/strong/realdata/` and feature a response $y \in \{0, 1\}$. We fit both least squares and logistic regression models to the data sets and examine the effect of the level of coarseness in the path by varying the length of the path ($l = 20, 50, 100$).

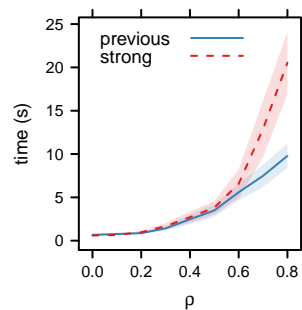

Figure 4: Time taken to fit a regularization path of SLOPE for least squares regression using either the strong or previous set algorithm.

There were no violations in any of the fits. The screening rule offers substantial reductions in problem size (Figure 5), particularly for the path length of 100, for which the size of the screened set of predictors ranges from roughly 1.5–4 times the size of the active set.

### 3.3.2 Performance

In this section, we introduce three new data sets: *e2006-tfidf* [34], *physician* [35], and *news20* [36]. *e2006-tfidf* was collected from Frandi [34], *news20* from `https://www.csie.ntu.edu.tw/~cjlin/libsvmtools/datasets` [37], and *physician* from `https://www.jstatsoft.org/`

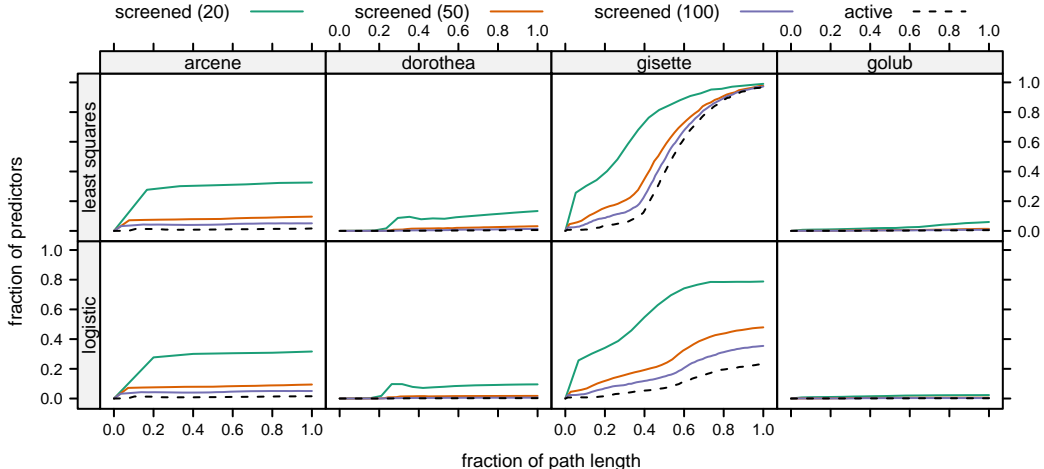

Figure 5: Proportion of predictors included in the model by the strong screening rule as a proportion of the total number of active predictors in the model for a path of $\lambda$ sequences. Three types of paths have are examined, using path lengths of 20, 50, and 100.

Table 1: Benchmarks measuring wall-clock time for four data sets fit with different models using either the strong screening rule or no rule.

| dataset | model | $n$ | $p$ | time (s) | |
|---|---|---|---|---|---|
| | | | | no screening | screening |
| dorothea | logistic | 800 | 88119 | 914 | 14 |
| e2006-tfidf | least squares | 3308 | 150358 | 43353 | 4944 |
| news20 | multinomial | 1000 | 62061 | 5485 | 517 |
| physician | poisson | 4406 | 25 | 34 | 34 |

`article/view/v027i08` [38]. We use the test set for *e2006-tfidf* and a subset of 1000 observations from the training set for *news20*.

In Table 1, we summarize the results from fitting sorted $\ell_1$-regularized least squares, logistic, Poisson, and multinomial regression to the four data sets. Once again, we see that the screening rule improves performance in the high-dimensional regime and presents no noticeable drawback even when $n > p$.

## 4   Conclusions

In this paper, we have developed a heuristic predictor screening rule for SLOPE and shown that it is a generalization of the strong rule for the lasso. We have demonstrated that it offers dramatic improvements in the $p \gg n$ regime, often reducing the time required to fit the full regularization path for SLOPE by orders of magnitude, as well as imposing little-to-no cost when $p < n$. At the time of this publication, an efficient implementation of the screening rule is available in the R package `SLOPE` [28].

The performance of the rule is demonstrably weaker when predictors in the design matrix are heavily correlated. This issue may be mitigated by the use of the previous set strategy that we have investigated here; part of the problem, however, is related to the clustering behavior that SLOPE exhibits: large portions of the total number of predictors often enter the model in a few clusters when regularization is strong. A possible avenue for future research might therefore be to investigate if screening rules for this clustering behavior might be developed and utilized to further enhance performance in estimating SLOPE.

## Broader Impact

The predictor screening rules introduced in this article allow for a substantial improvement of the speed of SLOPE. This facilitates application of SLOPE to the identification of important predictors in huge data bases, such as collections of whole genome genotypes in Genome Wide Association Studies. It also paves the way for the implementation of cross-validation techniques and improved efficiency of the Adaptive Bayesian version SLOPE (ABSLOPE [39]), which requires multiple iterations of the SLOPE algorithm. Adaptive SLOPE bridges Bayesian and the frequentist methodology and enables good predictive models with FDR control in the presence of many hyper-parameters or missing data. Thus it addresses the problem of false discoveries and lack of replicability in a variety of important problems, including medical and genetic studies.

In general, the improved efficiency resulting from the predictor screening rules will make the SLOPE family of models (SLOPE [3], grpSLOPE [6], and ABSLOPE) accessible to a broader audience, enabling researchers and other parties to fit SLOPE models with improved efficiency. The time required to apply these models will be reduced and, in some cases, data sets that were otherwise too large to be analyzed without access to dedicated high-performance computing clusters can be tackled even with modest computational means.

We can think of no way by which these screening rules may put anyone at disadvantage. The methods we outline here do not in any way affect the model itself (other than boosting its performance) and can therefore only be of benefit. For the same reason, we do not believe that the strong rules for SLOPE introduces any ethical issues, biases, or negative societal consequences. In contrast, it is in fact possible that the reverse is true given that SLOPE serves as an alternative to, for instance, the lasso, and has superior model selection properties [10, 39] and lower bias [39].

## Acknowledgments and Disclosure of Funding

We would like to thank Patrick Tardivel (Institut de Mathématiques de Bourgogne) and Yvette Baurne (Department of Statistics, Lund University) for their insightful comments on the paper. The computations were enabled by resources provided by the Swedish National Infrastructure for Computing (SNIC) at Lunarc partially funded by the Swedish Research Council through grant agreement no. 2018-05973.

The research of Małgorzata Bogdan was supported by the grant of the Polish National Center of Science no. 2016/23/B/ST1/00454. The research of Jonas Wallin was supported by the Swedish Research Council, grant no. 2018-01726. These entities, however, have in no way influenced the work on this paper.

## Footnotes

[1]The active set is then retrieved by sub-setting the first $k$ elements of the ordering permutation.

[2]Bogdan et al. [3] also presented a method called the *Gaussian* sequence that is a modification of the BH method, but it is not appropriate for our problems since it reduces to the lasso in the $p \gg n$ context.

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
