[Supplementary Material 1]

# Supplement to "The Strong Screening Rule for SLOPE"

Johan Larsson[1,*], Małgorzata Bogdan[1,2], and Jonas Wallin[1]

[1]*Department of Statistics, Lund University*
[2]*Department of Mathematics, University of Wroclaw*
[*]*Corresponding author:* *johan.larsson@stat.lu.se*

October 21, 2020

## 1 Proofs

### 1.1 Proof of Theorem 1

By definition, the subdifferential $\partial J(\beta; \lambda)$ is the set of all $g \in \mathbb{R}^p$ such that

$$J(y; \lambda) \geq J(\beta; \lambda) + g^T(y - \beta) = \sum_{j=1}^{p} |\beta|_{(j)} \lambda_j + g^T(y - \beta), \qquad (1)$$

for all $y \in \mathbb{R}^p$.

Assume that we have $K$ clusters $\mathcal{A}_1, \mathcal{A}_2, \ldots, \mathcal{A}_K$ (as defined per Equation 2 (main article)) and that $\beta = |\beta|_\downarrow$, which means we can rewrite (1) as

$$
\begin{aligned}
0 &\geq J(\beta; \lambda) - J(y; \lambda) + g^T(y - \beta) \\
&= \sum_{i \in \mathcal{A}_1} (\lambda_i |\beta|_{(i)} - g_i \beta_i - \lambda_i |y|_{(i)} + g_i y_i) + \ldots \\
&\quad + \sum_{i \in \mathcal{A}_K} (\lambda_i |\beta|_{(i)} - g_i \beta_i - \lambda_i |y|_{(i)} + g_i y_i).
\end{aligned}
$$

Notice that we must have $\sum_{i \in \mathcal{A}_j} (\lambda_i |\beta|_{(i)} - g_i \beta_i - \lambda_i |y|_{(i)} + g_i y_i) \leq 0$ for all $j \in \{1, 2, \ldots, K\}$ since otherwise the inequality breaks by selecting $y_i = \beta_i$ for $i \in \mathcal{A}_j^c$. This means that it is sufficient to restrict attention to a single set as well as take this to be the set $\mathcal{A}_i = \{1, \ldots, p\}$.

*Case* 1 ($\beta = \mathbf{0}$). In this case (1) reduces to $J(y; \lambda) \geq g^T y$. Now take a $c \in \mathcal{Z}$ where

$$\mathcal{Z} = \left\{ s \in \mathbb{R}^p \mid \operatorname{cumsum}(|s|_\downarrow - \lambda) \preceq \mathbf{0} \right\} \qquad (2)$$

and assume that $|c_1| \geq \cdots \geq |c_p|$ without loss of generality.

Clearly, $J(y; \lambda) \geq c^T y$ holds if and only if $J(y^*; \lambda) - c^T y^* \geq 0$ where

$$y^* = \arg\min_y \left\{ J(y; \lambda) - c^T y \right\}.$$

Now, since $J(y; \lambda)$ is invariant to changes in signs and permutation of $y$, it follows from the rearrangement inequality [HLP52, Theorem 368] that $|y|_1^* \geq \cdots \geq |y|_p^*$. This permits us to formulate the following equivalent problem:

$$\begin{aligned}
\text{minimize} \quad & y^T (\text{sign}(y) \odot \lambda - c) \\
\text{subject to} \quad & \text{sign}(y) = \text{sign}(c), \\
& |y_1| \geq \cdots \geq |y_p|.
\end{aligned}$$

To minimize the objective $y^T (\text{sign}(y) \odot \lambda - |c|) = |y|^T (\lambda - |c|)$, recognize first that we must have $y_1^* = y_2^*$ since $c \in \mathcal{Z}$, which implies $\lambda_1 - |c_1| \geq 0$. Likewise, $y_2^*(\lambda_1 - |c_1|) + y_2^*(\lambda_2 - |c_2|) \geq 0$ since $\lambda_1 + \lambda_2 - (|c_1| + |c_2|) \geq 0$, which leads us to conclude that $y_2^* = y_3^*$. Then, proceeding inductively, it is easy to see that $y_p^* \sum_{i=1}^{p}(\lambda_i - |c_i|) \geq 0$, which implies $y_1^* = \cdots = y_p^* = 0$. At this point, we have shown that $c \in \mathcal{Z} \implies c \in \partial J(\beta; \lambda)$.

For the next part note that $g \in \mathcal{Z}$ is equivalent to requiring $|g|_{(1)} \leq \lambda_1$ and

$$|g|_{(i)} \leq \sum_{j=1}^{i} \lambda_j - \sum_{j=2}^{i} |g|_{(j)}, \qquad i = 1, \ldots, p. \tag{3}$$

Now assume that there is a $c$ such that $c \in \partial J(\beta; \lambda)$ and $c \notin \mathcal{Z}$. Then there exists an $\varepsilon > 0$ and $i \in \{1, 2, \ldots, p\}$ such that

$$|c|_{(i)} \leq \sum_{j=1}^{i} \lambda_j - \sum_{j=2}^{i} |c|_{(j)} + \varepsilon, \qquad i = 1, \ldots, p.$$

Yet if $c = [\lambda_1, \ldots, \lambda_{i-1}, \lambda_i + \varepsilon, \lambda_{i+1}, \ldots, \lambda_p]^T$ then (1) breaks for $y = \mathbf{1}$, which implies that $c \notin \mathcal{Z} \implies c \notin \partial J(\beta; \lambda)$.

*Case* 2 ($\beta \neq \mathbf{0}$). Now let $|\beta_i| := \alpha$ for all $i = 1, \ldots, p$, since by construction all $\beta$ are equal in absolute value. Now (1) reduces to

$$\begin{aligned}
J(y; \lambda) &\geq J(\beta; \lambda) - g^T \beta + g^T y \\
&= \sum_{i=1}^{p} \lambda_i \alpha - \sum_{i=1}^{p} g_i \, \text{sign}(\beta_i) \alpha + g^T y \\
&= \alpha \sum_{i=1}^{p} (\lambda_i - g_i \, \text{sign}(\beta_i)) + g^T y.
\end{aligned} \tag{4}$$

The first term on the right-hand side of the last equality must be zero since otherwise the inequality breaks for $y = \mathbf{0}$. In addition, it must also hold that $\text{sign}(\beta_i) = \text{sign}(g_i)$ for all $i$ such that $|\beta_i| > 0$. To show this, suppose the opposite

is true, that is, there exists at least one $j$ such that $\text{sign}(g_j) \neq \text{sign}(\beta_j)$. But then if we take $y_j = \alpha \, \text{sign}(g_j)$ and $y_i = -\alpha \, \text{sign}(g_i)$, (4) is violated, which proves the statement by contradiction.

Taken together, this means that we have $g \in \mathcal{H}$ where

$$\mathcal{H} = \left\{ s \in \mathbb{R}^p \mid \sum_{j=1}^p (|s_j| - \lambda_j) = 0. \right\}$$

We are now left with $J(y; \lambda) \geq g^T y$, but this is exactly the setting from case one. Direct application of the reasoning from that part shows that we must have $g \in \mathcal{Z}$. Connecting the dots, we finally conclude that $c \in \mathcal{Z} \cap \mathcal{H} \implies c \in \partial J(\beta; \lambda)$.

## 1.2 Proof of Proposition 1

Suppose that we have $\mathcal{B} \neq \varnothing$ after running Algorithm 1 (main article). In this case we have

$$\text{cumsum}(c_{\mathcal{B}} - \lambda_{\mathcal{B}}) = \text{cumsum} \left( \left( \left| \nabla f(\hat{\beta}(\lambda^{(m+1)})) \right|_{\downarrow} \right)_{\mathcal{B}} - \lambda_{\mathcal{B}}^{(m+1)} \right) \prec \mathbf{0},$$

which implies via Theorem 1 (main article) and Equation 3 (main article) that all predictors in $\mathcal{B}$ must be inactive and that $\mathcal{S}$ contains the true support set.

## 1.3 Proof of Proposition 2

We need to show that the strong rule approximation does not violate the inequality on the fourth line in Algorithm 1 (main article). Since $\text{cumsum}(y) \succeq \text{cumsum}(x)$ for all $x, y \in \mathbb{R}^p$ if and only iff $y \succeq x$, it suffices to show that

$$|c_j(\lambda^{(m)})| + \lambda_j^{(m)} - \lambda_j^{(m+1)} \geq |c_j(\lambda^{(m+1)})|$$

for all $j = 1, 2, \ldots, p$, which in turn means that Algorithm 1 (main article) with $|c_j(\lambda^{(m)})| + \lambda_j^{(m)} - \lambda_j^{(m+1)}$ as input cannot result in any violations.

From our assumptions we have

$$|c_j(\lambda^{(m+1)}) - c_j(\lambda^{(m)})| \leq |\lambda_j^{(m+1)} - \lambda_j^{(m)}|.$$

Using this fact, observe that

$$\begin{aligned} |c_j(\lambda^{(m+1)})| &\leq |c_j(\lambda^{(m+1)}) - c_j(\lambda^{(m)})| + |c_j(\lambda^{(m)})| \\ &\leq \lambda_j^{(m)} - \lambda_j^{(m+1)} + |c_j(\lambda^{(m)})|. \end{aligned}$$

## 1.4 Proof of Proposition 3

Let $c = (\nabla f(\hat{\beta}(\lambda)))$ and $\lambda_1 = \lambda_2$ and assume without loss of generality that $p = 2$ and $c_1 \geq c_2 \geq 0$. Recall that the strong rule for lasso discards the $j$th predictor whenever $c_j < \lambda_1$. There are three cases to consider.

*Case* 3 ($c_2 \leq c_1 < \lambda_1$). $\mathrm{cumsum}(c - \lambda) \prec 0$, which means both predictors are discarded.

*Case* 4 ($c_1 \geq \lambda_1 > c_2$). The first predictor is retained since $\mathrm{cumsum}(c - \lambda)_1 > 0$; the second is discarded because $c_2 \leq \lambda$.

*Case* 5 ($c_1 \geq c_2 \geq \lambda_1$). Both predictors are retained since $\mathrm{cumsum}(c - \lambda) \succeq 0$.

The two results are equivalent for the lasso and thus the strong rule for SLOPE is a generalization of the strong rule for the lasso.

## 2  Algorithms

---
**Algorithm 1** Strong set algorithm
---
$\mathcal{V} \leftarrow \varnothing$
$\mathcal{E} \leftarrow \mathcal{S}(\lambda^{(m+1)}) \cup \mathcal{T}(\lambda^{(m)})$
**do**
    compute $\hat{\beta}_{\mathcal{E}}(\lambda^{(m+1)})$
    $\mathcal{V} \leftarrow$ KKT violations in full set
    $\mathcal{E} \leftarrow \mathcal{E} \cup \mathcal{V}$
**while** $\mathcal{V} \neq \varnothing$
**return** $\hat{\beta}_{\mathcal{E}}(\lambda^{(m+1)})$

---

---
**Algorithm 2** Previous set algorithm
---
$\mathcal{V} \leftarrow \varnothing$
$\mathcal{E} \leftarrow \mathcal{T}(\lambda^{(m)})$
**do**
    compute $\hat{\beta}_{\mathcal{E}}(\lambda^{(m+1)})$
    $\mathcal{V} \leftarrow$ KKT violations in $\mathcal{S}(\lambda^{(m+1)})$
    **if** $\mathcal{V} = \varnothing$ **then**
        $\mathcal{V} \leftarrow$ KKT violations in full set
    **end if**
    $\mathcal{E} \leftarrow \mathcal{E} \cup \mathcal{V}$
**while** $\mathcal{V} \neq \varnothing$
**return** $\hat{\beta}_{\mathcal{E}}(\lambda^{(m+1)})$

---

## References

[1]  G.H. Hardy, J.E. Littlewood, and G. Pólya. *Inequalities*. Second. Cambridge, United Kingdom: Cambridge University Press, 1952. ISBN: 0-521-05206-8.



[Supplementary Material 2]



Legend (top): screened (20) — screened (50) — screened (100) — active

Y-axis: fraction of predictors

X-axis: fraction of path length

Column labels: arcene, dorothea, gisette, golub

Row labels: OLS, logistic

[Supplementary Material 3]



Legend: screening (blue), no screening (red)

Y-axis groups (top to bottom): multinom, poisson, logistic, OLS

Within each group, ρ values: 0.999, 0.99, 0.5, 0

X-axis: time (s) — 1, 10, 100