[Reviews · NeurIPS 2020]

Review 1

Summary and Contributions: The paper addresses the problem of efficiently computing regularization paths for the SLOPE regularized generalized linear models. The contribution consists of a new formulation of the subgradient of the the SLOPE subgradient and allows the justification of algorithm 1 (and its accelerated version algorithm 2) which provides a superset of the solution support based on the gradient of the loss. Given this result is proposed a 'strong rule' strategy following the seminal work from Tibshirani et al. and which is implemented in the GLMNET.

Strengths: The papers comes with extensive experimental results (a full family for generalized linear models) and some R code in supplementary material to replicate the experiments. This is well appreciated. Fast computation of a SLOPE paths is novel to my knowledge and contribution is relevant and timely.

Weaknesses: Major concerns: - My biggest concern is the poor scholar work done by the authors on the literature review. Many seminal papers on SLOPE-like regularizations are not cited: OSCAR (Bondell et al. 2007), OWL: Atomic Formulation, Projections, and Algorithms (Zeng et al.). Due to late upload on Arxiv this ICML 2020 paper (Bao et al. https://arxiv.org/abs/2006.16433) should also be added to the paper. Also, this later paper does a much better job at acknowledging previous research of screening rules for sparse GLMs. The strong rules have been further improved by many groups over the years, in particular with safe rules that provably will not discard features wrongly: El Ghaoui et al. 2012, Fercoq et al. 2015, Shibagaki et al. 2016, Ndiaye et al. 2017 just to name a few. There are also other theoritical results on SLOPE such as Bellec et al. 2018. To sum the list of references should be significantly complemented. - There are a few places in the text that lack final proof-reading: - L206: The sentence that explains data simulation is broken and the \rho is not defined here but rather L229. - Comparing solvers is not convincing without details on the convergence criteria that is used to stop the iterative solver (here FISTA). This needs to be clarified. - When you write '[...] this is lower than the cost of a single gradient step if a first-order method is used to compute the SLOPE solution.' I am confused as for me it's the exact same code for a proximal gradient step (gradient is nxp and then you have a PAV for the prox). - There are to me a few typos in the proof. On page 2 of supplementary pdf in the sentence that starts with 'To minimize the objective...' the vector c is not necessarily positive so the equality is not true. Please check if some absolute values are missing. In the next paragraph, when you write 'For the next part note that (2) is equivalent' I would write: 'For the next part note that $g \in \mathcal{Z}$ is equivalent' as (2) does not involve 'g' per se. Typos - L90 : the the - The name 'Malgorzata Bogdan' is typed differently in 2 references.

Correctness: correct AFAIK.

Clarity: yes pretty clear

Relation to Prior Work: many references are missing

Reproducibility: Yes

Additional Feedback: I have read the author's response. I consider that it addresses my comments and I will keep my grade at 7 (good submission / accept).


Review 2

Summary and Contributions: Screening rules for the Lasso allow to identify inactive predictors before or along the optimization process. They can lead to huge speedups when p >>n and the regularization is strong enough. The SLOPE estimator is a refinement of the lasso where the penalty used is a weighted L1, which performs better than the Lasso on correlated designs. The paper derives unsafe screening rules for the SLOPE estimator, thus improving the speed of numerical solvers. The contribution is an adapted reformulation of the subdifferential of the slope penalty, and the adaptation of Tibshirani et al Strong rule for the Lasso to this penalty.

Strengths: The work is well presented, motivated, application is interesting and speedups are demonstrated.

Weaknesses: Questions: - constructing a path of decreasing λ > can you detail more at this point ? Given the explosive cardinality of the set of possible lambdas, it seems many choices are possible. Do you fix the ratios between consecutive lambdas and only tune one parameter, etc? This is done L184, you could add a reference to 3.1 on L40. - Recall that we are attempting ... > I don't think you said that you were considering the sequential setting before (some screening rules work in the non sequential setting). this deserves a clarification : your rule is useful in the sequential setting but cannot be applied if the problem is to be solved with a single lambda (eg, fitting on full dataset at optimal lambda, after determining the optimal lambda bia K-fold CV) - the true solution is never available (ie in l 137 you wouldn't have the exact gradient at the previous value on the path). Can you comment ? - which is equivalent to breaking the assumption that the gradient vector is a piece-wise linear function > can you explain ? More than being piecewise linear, you need its slope to be bounded by 1, don't you ? - There is a rich literature on Screening rules that authors fail at citing in the introduction. Experimental questions: - you chose t = 10 −2 (going from lambda_max to lambda_max / 100). From my experience in cross validation settings, you often need to go to alpha_max / 1000. If you keep l = 100, this makes your gride coarser, hence the rule less efficient as the gradient approximation is less correct and the RHS in your screening rules can even become negative (no screening happening)? See [1], Fig 5b. I would like to see the efficiency of your rule in this setting. - The usefulness of the screening rule depends on the frequency by which it is violated. > in my opinion, it is better to have a violating screening rules which screens a lot of variables than an ineffective one being very conservative. This is at the heart of working set policies (see [2,3]) - You often use n = 200 and stop the path when the number of selected variables grow larger than n (is it possible? For the Lasso, there always exists a solution with support of size at most n). So your solutions are quite sparse, and screening remains effective However, because of the bias of the Lasso penalty, optimal solutions from a prediction performance (MSE on left out data) are usually quite dense. To be fully convincing, you may need to run your simulations with larger n, eg the rcv1_train, log1p, news20 of LIBSVM (https://www.csie.ntu.edu.tw/~cjlin/libsvmtools/datasets/) Lesser important questions: - l 74 can you explain how you break ties and why they don't matter ? - l 130 why should the screened set (eliminated variables) be included in the active set ? shouldn't the equation be "we risk of returning S not included in complement(T) ?" l128 your screened set sounded to me like the set of screened variables, ie removed variables; from the sequel I infer that it is in fact the set of variables who survived screening (since "this screened set should resemble the active set"). This should be clarified. If S is the set of selected variables, you should fear returning S not containing T, no ? - in the equation below l145 (if you can number all your equations when there is no horizontal room constraint, in my opinion it makes communication easier), the RHS is a vector. - L184 could plot an example of such sequence to ease understanding ? Reformulations (I don't expect answers if you don't have room): rules that allow predictors to be discarded before estimating the model > most efficient screening rules are dynamic, i.e. discard coefficients *along the optimization process*. I suggest using the latter formulation. Our rule is heuristic > could you consider using the standard terminology "unsafe" here ? Following the "Safe rules" paper by the Lasso, and follow ups in Theorem 1 wedge is not defined, using a plain "and" or '&' may be clearer for someone not familiar with this logic notation - active set can be ambiguous in the literature, you may want to define it as the support of the solution somewhere

Correctness: Yes

Clarity: Yes

Relation to Prior Work: Should be improved Refs: [1] Ndiaye et al, Gap Safe Screening Rules for Sparsity Enforcing Penalties, JMLR 2017 [2] Johnson and Guestrin, BLITZ : A Principled Meta-Algorithm for Scaling Sparse Optimization, ICML 2015 [3] Massias et al, Celer: a fast solver for te Lasso with dual extrapolation, ICML 2018. [4] El Ghaoui, et al, Safe feature elimination in sparse supervised learning. J. Pacific Optim., 8(4):667–698, 2012. [5] Fercoq et al, Mind the duality gap: safer rules for the lasso, ICML 2015 [6] Shibagaki et al, Simultaneous safe screening of features and samples in doubly sparse modeling, ICML 2016. [7] Bonnefoy et al, A dynamic screening principle for the Lasso

Reproducibility: Yes

Additional Feedback: REBUTTAL FEEDBACK: The authors have answered my concerns about limited validation with larger datasets where the speedups are still visible. They also say to have fixed lack of references to previous work. The contribution in the computation of the subgradient is novel enough as screening rules for non separable penalties are non trivial to my knowledge. I have increased my grade from 6 to 7


Review 3

Summary and Contributions: The authors are interested in the Sorted L-One Penalized Estimation (SLOPE) algorithm: a generalisation of LASSO, which however doesn’t yet enjoy the same numerical performances. They propose a screening rule - a rule for discarding predictors without estimating the model - for SLOPE, which they prove is a generalisation of the strong screening rule for LASSO. They then conduct numerical experiments and show the proposed rule behaves very well in practice, with substantial time gains when p>>n without inducing additional cost when p < n.

Strengths: The authors derive a screening rule for SLOPE, which is both novel and impactful, as SLOPE is known to have a better behaviour than LASSO in cases of correlated predictors. They focus on the sub differential of the SLOPE regularisation term, taking inspiration of how strong screening rules are derived for LASSO - and present a simple algorithm (Algorithm 1) outputting a superset of the predictors which have nonzero coefficients in the next iteration. They then propose a faster, more efficient version of Algorithm 1, which has the same outputs while only keeping a scalar into memory (instead of a cumsum vector), and enjoys a linear complexity in the size of the data. The authors show that the strong screening rule for SLOPE outputs the same “screened set” as the strong rule for LASSO if all components of vector \lambda are the same: this shows that the proposed rule is a strict generalisation of the proposed rule for LASSO. Experiment design and results are well explained. The time gains displayed in section 3.2 are particularly impressive.

Weaknesses: The authors point out that the gradient used as input of Algo1&2 must be approximated (line 126), and say that this suggests defining the strong rule for SLOPE with the values of c and \lambda given in line 137. This motivation is not clear to me, as it appears possible that the gradient estimate be arbitrarily far away from its real value (thus potentially leading to violations of the rule). As far as I understand, Proposition 2 and 3 (which state guarantees for the strong screening rule for SLOPE) concern Algorithm 1 with an input that depends on the true gradient ("c" given on line 137), which has been said (lines 125/126) not to be available in practice. Shouldn’t guarantees be studied for the actual input that are used in Algorithm 1 (i.e. gradient estimate)? Notably, the different scenarios for the violation of the rules mentioned in 2.2.3 do not consider the possibility of a terribly biased gradient estimation. Results shown in the experiment section suggest that violations happen very rarely nonetheless.

Correctness: As far as my understanding of this subject goes, the claims are correct, and generalise known claims for LASSO.

Clarity: Apart from the details concerning gradient availability and estimation (Section 2.2.2), the paper is well written and easy to follow.

Relation to Prior Work: Although there is no dedicated “Related works” section, the relation to prior work is described clearly in the introduction section. This helps the reader understand the positioning of the paper and the novelty of the claims.

Reproducibility: Yes

Additional Feedback: ---EDIT AFTER AUTHOR RESPONSE--- I would like to thank the authors for taking the time to clarify the points I mentioned. There was indeed a misunderstanding for my second remark. I would therefore like to raise my score from a 6 to a 7.


Review 4

Summary and Contributions: The authors propose a variant of the strong rule for Sorted L-One Penalized Estimation (SLOPE). The proposed approach identifies and removes the inactive predictors to improve the efficiency of solving SLOPE. Experiments show that the proposed method is effective on synthetic datasets.

Strengths: The strengths of the work are as follows. 1. This paper aims to derive a screening rule for SLOPE, which is important in sparsity learning. 2. The proposed screening rule is easy to implement.

Weaknesses: The weaknesses of the work are as follows. 1. The novelty is limited, as this paper is a simple extension of the strong rule [4] to SLOPE [1]. 2. The motivation is unclear. The authors extend an unsafe screening rule [4] to SLOPE, while there are some safe and efficient screening rules, such as GAP safe rules [2] and EDPP [3]. The authors may want to explain the advantages of the strong rule [4] for SLOPE. 3. Experiments on real datasets are insufficient. The authors conduct experiments on small datasets. Moreover, the proposed method only achieves significant performance improvement on the smallest dataset with n=36. [1] Ma lgorzata Bogdan, Ewout Van Den Berg, Chiara Sabatti, Weijie Su, and Emmanuel J Candes. SLOPE: adaptive variable selection via convex optimization. The annals of applied statistics, 9(3):1103, 2015. [2] Eugene Ndiaye, Olivier Fercoq, Alexandre Gramfort, and Joseph Salmon. Gap safe screening rules for sparsity enforcing penalties. The Journal of Machine Learning Research, 18(1):4671-4703, 2017. [3] Jie Wang, Jiayu Zhou, Peter Wonka, and Jieping Ye. Lasso screening rules via dual polytope projection. In Advances in neural information processing systems, pages 1070-1078, 2013. [4] Robert Tibshirani, Jacob Bien, Jerome Friedman, Trevor Hastie, Noah Simon, Jonathan Taylor, and Ryan J Tibshirani. Strong rules for discarding predictors in lasso-type problems. Journal of the Royal Statistical Society: Series B (Statistical Methodology), 74(2):245-266, 2012.

Correctness: Experiments are insufficient to support the authors' claims. The authors may want to conduct experiments on larger datasets to demonstrate the effectiveness of the proposed method.

Clarity: This paper is well written and easy to follow.

Relation to Prior Work: The authors may want to detail the differences between the proposed method and the strong rule [1]. [1] Robert Tibshirani, Jacob Bien, Jerome Friedman, Trevor Hastie, Noah Simon, Jonathan Taylor, and Ryan J Tibshirani. Strong rules for discarding predictors in lasso-type problems. Journal of the Royal Statistical Society: Series B (Statistical Methodology), 74(2):245-266, 2012.

Reproducibility: Yes

Additional Feedback: Post-rebuttal update I would like to thank the authors' responses and additional evaluation on large datasets. They have partially addressed my concerns about experiments. I thus would like to raise my score from 4 to 5. Unfortunately, my major concern about the novelty (contribution) remains the same. The reasons are as follows. 1. According to the rebuttal and other reviewers' comments, the new formulation of subgradients in Theorem 1 is the major contribution of this submission. The subsequent extension to strong rule is straightforward. I agree that the derivation of the subgradients is non-trivial. However, I am not convinced that it could make this submission above the bar of NeurIPS, especially compared to the NeurIPS19 paper [11]. 2. The authors mentioned that the NeurIPS19 paper [11] has found the subgradients of SLOPE. They need to explicitly point out that: a) why the subgradients in [11] are not suitable for an extension to strong rule; b) what are the advantages of the subgradients found by this work compared to that of [11].

[Author Response · NeurIPS 2020]

We thank all reviewers for their thoughtful comments and suggestions. We provide our feedback below.

**Reviewer 1** "The contribution is relevant and timely." Thank you for your encouraging comments. "Many seminal
papers on SLOPE-like regularizations are not cited." We acknowledge the need for a more extensive section on the
previous literature and will revise the introduction accordingly. "[...] details on the convergence" Convergence is
obtained when the duality gap as a fraction of the primal is less than $10^{-5}$ and the relative level of infeasiblity (see the
appendix of Bogdan et al. 2015 [3]) is less than $10^{-3}$. We will include this in the revision. Regarding the computational
cost of the rule, the rule sorts the gradient but does not solve the prox, which makes the cost slightly lower than the
cost of a gradient step. We will clarify this in the revision. Thank you for pointing the typos in the proof. They will be
corrected together with other observed inconsistencies.

**Reviewer 2** "The work is well presented, motivated, application is interesting and speedups are demonstrated." Thank
you for your positive feedback. "constructing a path of decreasing $\lambda$ [...]". We will provide the suggested reference.
"your rule is useful in the sequential setting but cannot be applied [...] with a single lambda" It is possible to use the
rule non-sequentially since the gradient for the null model is always available. We will clarify this in the revision. "the
true solution is never available [...]" Actually, our algorithm calculates the exact solution (to numerical precision)
at each step, so the exact gradients at the previous steps are known. Please note that at each step we check the KKT
conditions and, if needed, recalculate SLOPE after adding predictors removed by the rule. We observed that such
corrections are rarely needed in practice. "More than being piecewise linear, you need its slope to be bounded by 1,
don't you?" At the intervals where the path is linear the unit bound is trivially satisfied (see the similar reasoning
for the lasso). "References" Thank you for a large set of references. We will extend the review section accordingly.
"I would like to see the efficiency of your rule in this setting". We have extended the experiments to analyze the
effectiveness of the rule with different path lengths (and thus coarseness). For the arcene data set and OLS model,
for instance, the average proportion of eliminated predictors (out of the total) is 0.74, 0.92, 0.96 for path lengths of
20, 50, and 100 respectively. Section 3.3.1 will be updated and extended accordingly. "it is better to have a violating
[...]". We agree and will revise. "For the Lasso, there always exists a solution with support of size at most n".
This is not the case with SLOPE. Due to clustering, SLOPE can return even p nonzero coefficients (assuming some
of them are equal to each other in absolute value) (cf. `https://arxiv.org/abs/2004.09106`). "[...] you may
need to run your simulations with larger $n$". We have updated the results with new datasets (see the included table).

"can you explain how you break ties and why
they don't matter?" SLOPE clusters variables
and averages the penalty coefficients over the
ties (see Bogdan et al. [3] for SLOPE prox).
"why should the screened set [...]" We agree
and will clarify this passage in the revision.
"L184 could plot [...]" We agree but are re-
grettably not able to add such a plot due to

| dataset | model | $n$ | $p$ | time (s) | |
|---|---|---|---|---|---|
| | | | | no screening | screening |
| dorothea | logistic | 800 | 88119 | 845 | 14 |
| e2006 | OLS | 3308 | 150358 | 43335 | 4874 |
| news20 | multinomial | 200 | 62061 | 2101 | 133 |
| physician | poisson | 4406 | 25 | 34 | 34 |

space constraints. We will also take into account other editorial suggestions and include suggested references.

**Reviewer 3** "The authors derive a screening rule for SLOPE, which is both novel and impactful." Thank you; we
appreciate the supportive feedback. "[...] the gradient estimate be arbitrarily far away from its real value". This is true
in theory, but unlikely in practice. In fact it has been empirically shown both for the lasso and SLOPE that in most cases
the unit bound is conservative. In our article we show that violations are rare for typical data sets. "Shouldn't guarantees
be studied [...]" We think there might be a misunderstanding here. If Algorithm 1 is used with a true gradient than it
returns the true support. Our rule relies on replacing the true gradient with an estimate based on the unit bound and
Proposition 2 specifies conditions under which Algorithm 1 returns the superset of the support. "[...] biased gradient
estimation." Unless any of the mentioned events occurs the gradient is linear and bounded by the unit bound. Please
note that the gradient at the previous step is known.

**Reviewer 4** "This paper aims to derive a screening rule for SLOPE, which is important in sparsity learning [and] [...]
easy to implement." Thank you for the remarks. "The novelty is limited [...]" We respectfully disagree. Developing
screening rules for SLOPE is notoriously difficult due to the non-separability of the penalty; ours is nevertheless one
of the first attempts to do so. We are aware of only one article about a safe rule for SLOPE published in ICML (after
we submitted). "The motivation is unclear. [...]" It is not clear that it's possible to extend GAP safe and EDPP
rules to SLOPE. Since our submission, Bao et al. (`https://arxiv.org/abs/2006.16433`) have published a paper
describing the first safe rule for SLOPE. Yet due to the non-separability of the penalty, this rule requires iteratively
screening predictors during optimization, which means predictors cannot be screened prior to fitting, which we think
highlights the difficulty in developing screening rules for SLOPE. "Experiments on real datasets are insufficient." We
agree. See the included table. "The authors may want to detail the differences [...]." The difference between the lasso
and SLOPE is that SLOPE has a non-separable penalty, which leads to a more complicated subgradient. In this paper
we derived a form of the subgradient that enables us to efficiently generalize the strong rule for the lasso to SLOPE.

[Meta-Review · NeurIPS 2020]

The work is well presented, motivated. Applications are interesting and speedups are demonstrated for computing the Slope estimator. The contribution proceeds from a reformulation of the subdifferential of the Slope penalty, and the adaptation of Tibshirani et al. "Strong rule for the Lasso" to this penalty. To improve the general reading, I recommend clarifying the point raised by R#4: "what are the advantages of the subgradients found by this work compared to that of [11]" in the camera ready version.